# UWSOD: Toward Fully-Supervised-Level Capacity Weakly Supervised Object Detection

Yunhang Shen[1]     Rongrong Ji[1,2,3*]     Zhiwei Chen[1]     Yongjian Wu[4]     Feiyue Huang[4]

[1]Media Analytics and Computing Lab, Department of Artificial Intelligence
School of Informatics, Xiamen University, 361005, China
[2]Institute of Artificial Intelligence, Xiamen University, 361005, China
[3]Peng Cheng Laborotory, Shenzhen, China
[4]Tencent Youtu Lab, Shanghai, China
shenyunhang01@gmail.com     rrji@xmu.edu.cn     zhiweichen.xmu@gmail.com
littlekenwu@tencent.com     garyhuang@tencent.com

## Abstract

Weakly supervised object detection (WSOD) has attracted extensive research attention due to its great flexibility of exploiting large-scale dataset with only image-level annotations for detector training. Despite its great advance in recent years, WSOD still suffers limited performance, which is far below that of fully supervised object detection (FSOD). As most WSOD methods depend on object proposal algorithms to generate candidate regions and are also confronted with challenges like low-quality predicted bounding boxes and large scale variation. In this paper, we propose a unified WSOD framework, termed UWSOD, to develop a high-capacity general detection model with only image-level labels, which is self-contained and does not require external modules or additional supervision. To this end, we exploit three important components, *i.e.*, object proposal generation, bounding-box fine-tuning and scale-invariant features. First, we propose an anchor-based self-supervised proposal generator to hypothesize object locations, which is trained end-to-end with supervision created by UWSOD for both objectness classification and regression. Second, we develop a step-wise bounding-box fine-tuning to refine both detection scores and coordinates by progressively select high-confidence object proposals as positive samples, which bootstraps the quality of predicted bounding boxes. Third, we construct a multi-rate resampling pyramid to aggregate multi-scale contextual information, which is the first in-network feature hierarchy to handle scale variation in WSOD. Extensive experiments on PASCAL VOC and MS COCO show that the proposed UWSOD achieves competitive results with the state-of-the-art WSOD methods while not requiring external modules or additional supervision. Moreover, the upper-bound performance of UWSOD with class-agnostic ground-truth bounding boxes approaches Faster R-CNN, which demonstrates UWSOD has fully-supervised-level capacity. The code is available at: `https://github.com/shenyunhang/UWSOD`.

## 1   Introduction

Different from fully supervised object detection (FSOD) [1–4] that requires bounding-box-level annotations, weakly supervised object detection (WSOD) [5–8] only needs image-level labels, which indicate the presence or absence of an object category. More recently, WSOD has attracted extensive attention to reducing the manual labelling effort to learn detectors. Unfortunately, due to the lack of

---

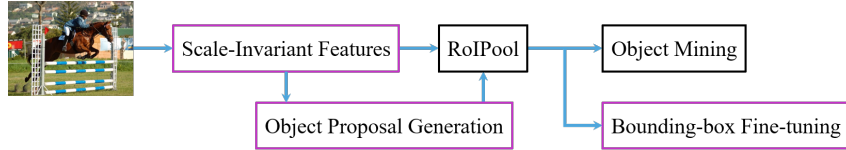

Figure 1: The overall flowchart of the proposed UWSOD. The full-image scale-invariant feature maps are first computed. Then object proposal generation provides candidate boxes to extract proposal features in RoIPool layer. Finally, the object mining phase outputs initial detection scores, and bounding-box fine-tuning phase further refines both scores and coordinates of proposals.

instance-level annotations, WSOD is still challenging to obtain satisfactory performance. Therefore, there is still a huge performance gap between WSOD and FSOD methods.

The recent widely-used paradigm for WSOD is a two-phase learning procedure, *i.e.*, object mining and instance refinement. In the first phase, multiple instance learning is employed to mine object from a set of candidate regions generated by object proposal algorithms, *e.g*., selective search [9] and edge boxes [10]. Then multiple parallel branches of instance refinement are trained to refine bounding boxes by using the preceding predictions as supervision. Although the above paradigm has achieved promising results, there still exist three major challenges: First, object proposal algorithms are adopted as external modules independent of the detectors. Such a multi-stage system hurts the detection accuracy and efficiency. Recent attempts in [11–13] learn to generate object proposals for WSOD. However, those methods are still not in an end-to-end fashion and require traditional object proposals [9, 10], motion segmentation [14] and additional video dataset [15, 16]. Second, the predicted bounding boxes may not cover object well, as they heavily relies on the quality of candidate boxes generated by object proposal algorithms, which limits further improvement with large margins. One way to reduce mislocalizations is to using bounding-box regression [17–20]. However, methods in [17–19] require super-pixel evidence and additional supervision to fine-tune bounding boxes, and regression module in [20] failed to consider the trade-off between the precision and recall requirements in different branches of instance refinement. Third, existing WSOD methods use multi-scale image pyramids [21] to remedy the scale-variation problem. However, they neglect the in-network feature hierarchy to handle large scale variations. And the increase of inference time and memory consumption makes the image pyramid infeasible for practical applications.

In this paper, we propose a unified WSOD framework, termed **UWSOD**, to develop a high-capacity general detection model with only image-level labels, which is self-contained and does not require external modules or additional supervision. In particular, we exploit three important components, *i.e.*, object proposal generation, bounding-box fine-tuning and scale-invariant features to address above challenges, as illustrated in Fig. 1. First, we propose an anchor-based self-supervised object proposal generator (SSOPG) to hypothesize object locations, which is trained end-to-end with supervision created by UWSOD for both objectness classification and regression. Second, to reduce mislocalizations, we propose a step-wise bounding-box fine-tuning (SWBBFT) to refine both detection scores and coordinates by progressively select high-confidence object proposals as positive samples, which bootstraps the quality of predicted bounding boxes. Third, we construct a multi-rate resampling pyramid (MRRP) to aggregate multi-scale contextual information, which is the first in-network feature hierarchy to handle scale variation in WSOD. Different to common FSOD that attach new parameter to build feature pyramids [22], MRRP does not need to learn new parameters and avoids over-fitting by sharing the same parameters of pre-trained backbones.

The contributions of this work are concluded as follows:

- We propose a unified weakly supervised object detection (UWSOD) framework, which is self-contained and does not require external modules or additional supervision to develop a high-capacity general detection model with only image-level labels.

- An anchor-based self-supervised proposal generator is proposed to hypothesize candidate object locations, which is end-to-end trainable with supervision created by UWSOD.

- We propose a step-wise bounding-box fine-tuning to refine both detection scores and coordinates progressively, which aims to bootstrap the quality of predicted bounding boxes.

- A multi-rates resampling pyramid is constructed to aggregate multi-scale contextual information, which is the first in-network feature hierarchy to handle scale variation in WSOD.

Extensive experiments on PASCAL VOC and MS COCO show that the proposed UWSOD achieves competitive results with the state-of-the-art methods while not requiring external modules or additional supervision. A crucial property of our model is that even with class-agnostic ground-truth bounding boxes, the upper-bound performance of UWSOD approach Faster R-CNN [2], thus having fully-supervised-level capacity.

## 2 Related work

### 2.1 Weakly supervised object detection

Recent widely-used WSOD learning procedure has two phases: object mining and instance refinement. The object mining phase is formulated as multiple instance learning to implicitly model latent object locations with image-level labels, which alternates between localizing object instances and training an appearance model [23–28]. Bilen *et al*. [7] selected proposals by parallel detection and classification branches in deep convolutional networks. Contextual information [29], attention mechanism [30], gradient map [31] and semantic segmentation [32] are leveraged to learn outstanding object proposals.

The instance refinement phase aims at explicitly learning the object location by making use of the predictions from the object mining phase. The top-scoring proposals from the preceding predictions are used as supervision to train the instance refinement classifier [33, 34, 8, 35]. Other different strategies [36–39] are also proposed to generate pseudo-ground-truth boxes and assign labels to proposals. Some methods exploit to improve the optimization of the overall framework that jointly learn the two-phase modules with min-entropy prior [40, 41], multi-view learning [42], continuation MIL [43], utilizing uncertainty [44–46] and generative adversarial learning [47]. Collaboration mechanisms are also exploited to take advantages of the complementary interpretations of weakly supervised tasks [48, 49] and different WSOD models [50]. With the output of the above two-phase paradigm, a separated fully-supervised detector can also be trained. Thus, many efforts [51, 52] have been made to mine high-quality bounding boxes for FSOD.

Some work also used additional annotations and data to improve the performance, *e.g*., object-size estimation [53], instance-count annotations [17], video-motion cue [54, 12] and human verification [55]. Knowledge transfer for cross-domain adaptation has been exploited, *e.g*., data adaption [56] and task adaption [57]. Methods in [19, 58, 59] trained object detection systems from different supervisions.

### 2.2 Object proposal generation

Object proposal methods aim to generate candidate object regions for an ensuing detector or segmentation model. Traditional methods include those based on grouping super-pixels, *e.g*., selective search [9] and multiscale combinatorial grouping [60], and those based on sliding windows, *e.g*., edge boxes [10]. In most existing WSOD, object proposal methods were typically adopted as external modules independent of the detectors. Few literature exploits trainable object proposal generation in WSOD. Cheng *et al*. [13] combined selective search [9] and a gradient-weighted class activation map [61] to generate more proposals. Tang *et al*. [11] used a smaller WSOD network [8] to refine the coarse proposals generated by edge boxes [10] on edge-like response maps. Singh *et al*. [12] used motion information in weakly-labelled videos to learn object proposals. However, all of they are not in an end-to-end fashion and still need traditional object proposals [9, 10], motion segmentation [14] and additional video dataset [15, 16]. In this work, the proposed anchor-based self-supervised proposal generator is end-to-end trainable and does not use external modules or additional information.

There are some WSOD methods focused on proposal-free paradigms by taking advantages of deep feature maps [62], class activation maps [63, 64] and generative adversarial learning [65]. However, such paradigm seriously depends on the quality of feature maps and is hard to distinguish different instances in challenging scenes.

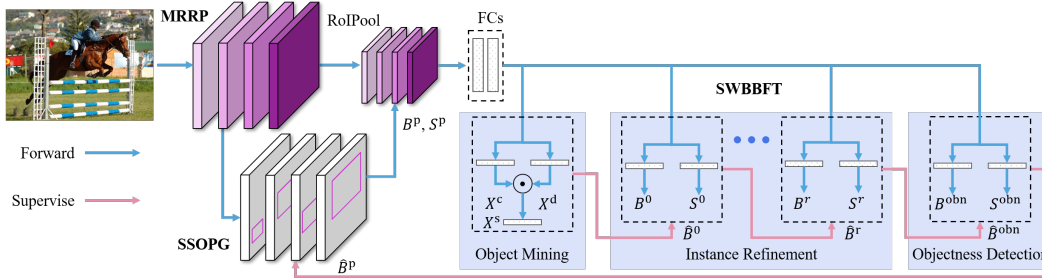

Figure 2: The figure illustrates the architecture of UWSOD.

## 2.3 Bounding-box regression

Bounding-box regression proposed in [1] framed localization as a regression problem, which is widely adopted by FSOD. However, only few work introduces bounding-box regression into WSOD due to the lack of instance-level annotations. Gao *et al*. [17] and Fang *et al*. [19] trained bounding box regressors with counting information or instance-level labels. Zeng *et al*. [18] combined superpixels straddling [66] and predicted scores to determine bounding-box regression targets. Yang *et al*. [67] leveraged category labels and action labels as location cues. Yanga *et al*. [35] introduced additional detection branch to jointly optimized the region classification and regression. Ren *et al*. [20] selected multiple pseudo boxes with non-maximum suppression for regression training. However, the above methods either require external modules and additional supervision, or fail to consider the trade-off between the precision and recall requirements in different refinement branches. In this work, the proposed step-wise bounding-box fine-tuning refines both classification and regression progressively.

## 2.4 Handling scale variation

Most WSOD methods heavily use image pyramids [21] to detect object across scales during training and testing to remedy the scale-variation problem. However, image pyramid method increases the inference time and neglects the in-network feature hierarchy to handle large scale variation. SNIP [68, 69] proposed a scale normalization strategy for FSOD, which selectively trains the objects of appropriate sizes in each image scale. However, SNIP is not adaptable to WSOD due to the lack of instance-level annotations. Another stream of utilizing multi-scale information in fully supervised learning is to consider both low- and high-level information. For example, encode-decoder structure in U-Net [70] and FPN [22] attaches a top-down pyramid-like structure to propagate information from top to bottom layers. However, it requires to attache new layers to build feature pyramids and may converge to an undesirable local minimum in WSOD. R-SSD [71] and RRC [72] gathered both low- and high-level feature maps, which cost more computational resource significantly.

## 3 The proposed method

### 3.1 Unified weakly supervised object detection (UWSOD) framework

In this paper, we introduce a unified weakly supervised object detection framework, which consists of three major components: self-supervised object proposal generator (SSOPG), step-wise bounding-box fine-tuning (SWBBFT) and multi-rate resampling pyramid (MRRP). The overall architecture of the proposed network is shown in Fig. 2. Given an input image, scale-invariant full-image feature maps are first extracted from the backbone with MRRP (Sec. 3.4). Then SSOPG (Sec. 3.2) predicts a set of high-confidence object proposals, which is followed by RoIPool layer to generate proposal features. Finally, the object mining phase outputs initial detection scores, and multi-branch SWBBFT refines both the scores and coordinates of proposals progressively to bootstrap the quality of predicted bounding boxes (Sec. 3.3). The overall loss function of UWSOD is:

$$L = L_{\text{SSOPG}} + L_{\text{OM}} + L_{\text{SWBBFT}}, \tag{1}$$

where $L_{\text{SSOPG}}$ and $L_{\text{SWBBFT}}$ are the loss functions of the proposed SSOPG and SWBBFT, which will discuss in the remainder of this section. And $L_{\text{OM}}$ is the loss function of object mining phase.Object mining phase [7, 29] forks the proposal features into two streams, *i.e.*, classification stream and detection stream, producing two score matrices $X^c, X^d \in \mathbb{R}^{R \times C}$ by two fully-connected layers, respectively. Both score matrices are normalized by softmax functions $\sigma(\cdot)$ over categories and proposals, respectively. Then the element-wise product of the output of the two streams is again

a score matrix: $X^{\mathrm{s}} = \sigma(X^{\mathrm{c}}) \odot \sigma(X^{\mathrm{d}})$. To acquire image-level classification scores, a sum pooling is applied: $\mathbf{y}_k = \sum_{r=1}^{R} X_{rk}^{\mathrm{s}}$. Then we obtain a cross-entropy loss function $\mathcal{L}_{\mathrm{OM}}$:

$$L_{\mathrm{OM}} = \sum_{i=1}^{n^{\mathrm{c}}} \Big\{ \mathbf{t}_i \log \mathbf{y}_i + (1 - \mathbf{t}_i) \log(1 - \mathbf{y}_i) \Big\}, \tag{2}$$

where $\mathbf{t} \in \{0, 1\}^{n^{\mathrm{c}}}$ is the image-level one-hot labels, $n^{\mathrm{c}}$ is the number of categories, and $\mathbf{t}_i$ is the ground-truth labels of whether an object of category $i$ is presented in the image.

## 3.2 Self-supervised object proposal generator (SSOPG)

We propose an anchor-based self-supervised object proposal generator (SSOPG), which takes full-image feature maps as input and outputs a set of rectangular object proposals, each with an objectness score. Anchors are regression references and classification candidates to predict object proposals. Generating anchors with the sliding window manner in feature maps has been widely adopted by anchor-based various detectors [2–4]. SSOPG use a small fully convolutional network to map each sliding window anchor to a low-dimensional feature, as in [2]. To this end, SSOPG has a $3 \times 3$ convolutional layer with 256 channels followed by two sibling $1 \times 1$ convolutional layers for objectness classification and regression, respectively. Formally, we denote $n^{\mathrm{a}}$ as the number of anchors in each location, and $h$ and $w$ as the height and width of feature maps, respectively. Thus, the regression layer has $4n^{\mathrm{a}}$ outputs encoding the coordinates of $n^{\mathrm{a}}$ boxes, and the objectness layer outputs $n^{\mathrm{a}}$ scores that estimate the probability of object for each proposal. Given a feature maps with spatial size $(h \times w)$, SSOPG outputs object proposals $B^{\mathrm{p}} \in \Re^{n^{\mathrm{a}} hw \times 4}$ and objectness scores $S^{\mathrm{p}} \in \Re^{n^{\mathrm{a}} hw \times 1}$. For inference, we apply non-maximum suppression (NMS) on $B^{\mathrm{p}}$ and pass the top-$n_{\mathrm{infer}}^{\mathrm{p}}$ object proposals $B_{\mathrm{infer}}^{\mathrm{p}}$ to the RoIPooling layers.

We leverage self-supervised learning to train SSOPG with supervision created by UWSOD without additional human effort or external modules. Our intuition is that as WSOD is able to discover category-specific object, it also has the potential to learn objectness instances. To this end, we attach a new objectness detection branch to the proposal features with two sibling fully-connected layers for objectness classification and regression, respectively. Given the output from instance refinement phase, i.e., bounding boxes $B^{\mathrm{r}} \in \Re^{n_{\mathrm{infer}}^{\mathrm{p}} \times n^{\mathrm{c}} \times 4}$ and scores $S^{\mathrm{r}} \in \Re^{n_{\mathrm{infer}}^{\mathrm{p}} \times n^{\mathrm{c}}}$, we generate pseudo-ground-truth objectness boxes $\hat{B}^{\mathrm{obn}} = \{B_{ij}^{\mathrm{r}} | i = \arg\max_i S_{ij}^{\mathrm{r}}, j = \{k | \mathbf{t}_k = 1\}\}$, where $B_{ij}^{\mathrm{r}}$ and $S_{ij}^{\mathrm{r}}$ denote the $i^{\mathrm{th}}$ predicted bounding box and score for $j^{\mathrm{th}}$ category. With $\hat{B}^{\mathrm{obn}}$, we label object proposals $B_{\mathrm{infer}}^{p}$ with an IoU ratio of $\lambda^{\mathrm{obn}}$ and sample $n_{\mathrm{train}}^{\mathrm{obn}}$ positive and negative training proposals, respectively. Finally, we select the top-$k^{\mathrm{obn}}$ predicted boxes from objectness detection branch as pseudo-ground-truth proposal boxes $\hat{B}^{\mathrm{p}}$ to supervise the above proposal generator, where $k^{\mathrm{obn}} = |\{\mathbf{t}_i | \mathbf{t}_i = 1\}|$. We label object proposals $B^p$ by an IoU ratio of $\lambda^{\mathrm{p}}$ with $\hat{B}^{\mathrm{p}}$ and sample $n_{\mathrm{train}}^{\mathrm{p}}$ positive and negative training proposals, respectively. The overall loss function of SSOPG is:

$$\begin{aligned} L_{\mathrm{SSOPG}} = &\sum_i L_{\mathrm{BCE}}(S_i^{\mathrm{obn}}, T_i^{\mathrm{obn}}) + \sum_i L_{\mathrm{smoothL1}}(B_i^{\mathrm{obn}}, \bar{B}_i^{\mathrm{obn}}) \\ &+ \sum_i L_{\mathrm{BCE}}(S_i^{\mathrm{p}}, T_i^{\mathrm{p}}) + \sum_i L_{\mathrm{smoothL1}}(B_i^{\mathrm{p}}, \bar{B}_i^{\mathrm{p}}) \end{aligned}, \tag{3}$$

where $T_i^{\mathrm{obn}}$ and $\bar{B}_i^{\mathrm{obn}}$ are the classification and regression targets of the $i^{\mathrm{th}}$ object proposal for objecness detection branch, and $T_i^{\mathrm{p}}$ and $\bar{B}_i^{\mathrm{p}}$ are the targets for object proposal generator. $L_{\mathrm{BCE}}$ is the binary sigmoid cross-entropy loss and $L_{\mathrm{smoothL1}}$ is the smooth $L_1$ loss as with [2].

## 3.3 Step-wise bounding-box fine-tune (SWBBFT)

Most state-of-the-art WSOD methods only apply classifier refinement [8] to rescore object proposals, which may result in low-quality predicted bounding boxes. As they heavily rely on the quality of candidate boxes generated by object proposal algorithms, which limit further improvement with large margins. Although recent methods in [17–20] integrate bounding-box regression in WSOD, they either require external modules or additional supervision, and fail to consider the trade-off between the precision and recall requirements in different refinement branches.

To reduce mislocalizations, we propose step-wise bounding-box fine-tuning (SWBBFT), which progressively selects high-confidence object proposals as positive samples to refine both detection scores and coordinates. Our intuition is that the former branches in instance refinement have large ambiguity of selecting positive and negative samples, as their pseudo-ground-truth labels are noisy and do not cover object well. Thus, we step-wisely learn instance refinement from low-quality to high-quality positive samples. To this end, based on vanilla classifier refinement [8], we first add bounding-box regression to each branch, which enables fine-tune both scores and coordinates in all refinement branches. We use a series of IoU thresholds $\lambda^{\mathrm{f}} = \{\lambda_i^{\mathrm{f}}, \ldots, \lambda_{n^{\mathrm{f}}}^{\mathrm{f}}\}$ to label positive and negative proposals and optimize each branch at separate IoU level, where $n^{\mathrm{f}}$ is the number of refinement branches. Thus, the corresponding loss function is:

$$L_{\mathrm{SWBBFT}} = \sum_{r=1}^{n^{\mathrm{f}}} \big( \sum_i \mathbf{y}_{T_i^{\mathrm{r}}} L_{\mathrm{CE}}(S_i^{\mathrm{r}}, T_i^{\mathrm{r}}) + \sum_i L_{\mathrm{smoothL1}}(B_i^{\mathrm{r}}, \bar{B}_i^{\mathrm{r}}) \big), \qquad (4)$$

where $T_i^{\mathrm{r}}$ and $\bar{B}_i^{\mathrm{r}}$ are the classification and regression targets for the $i^{\mathrm{th}}$ object proposal in the $r^{\mathrm{th}}$ branch, respectively, and $L_{\mathrm{CE}}$ is the softmax cross-entropy loss. To acquire training targets for each branch, we directly use the highest-score detection results from the preceding predictions as pseudo-ground-truth bounding boxes [8].

We restrict $\lambda^{\mathrm{f}}$ to be in descending order, *i.e.*, $\{\lambda_i^{\mathrm{f}} \leq \cdots \leq \lambda_{n^{\mathrm{f}}}^{\mathrm{f}}\}$. The descending order $\lambda^{\mathrm{f}}$ offers a good trade-off between the precision and recall requirements in different refinement branches. As the former branches establish a high-recall set of positive samples, while the successive branches receive high-precision positive samples. Step-wise fashion guarantees a sequence of effective refinement branches of increasing quality. As the set of positive samples decreases quickly with $\lambda^{\mathrm{f}}$, we sample all branches to guarantee that they have a fixed proportion for positive and negative samples.

### 3.4 Multi-rate resampling pyramid (MRRP)

Aggregating multi-scale information is critical for detectors to exploit context and achieve better performance in challenging conditions. Existing WSOD methods leverage multi-scale image pyramids [21] to remedy the scale-variation problem. However, it neglects the in-network feature hierarchy to handle large variation of scale.

Inspired by spatial pyramid pooling [73] and its successful variances in fully supervised learning [74–77], we construct a multi-rate resampling pyramid (MRRP) to aggregate multi-scale contextual information, in which each level shares the same parameters. Our intuition is that integrating information from other receptive field helps widen the scales, thus it can alleviate such ambiguities and reduce information uncertainty in the local area. Thus, we use a large range of receptive fields to describe objects at different scales. Regarding current backbone models, they commonly set receptive fields at the same size with a regular sampling grid on a feature map. Therefore, we generalize trident block [77] to iteratively replicates $n^{\mathrm{m}}$ parallel streams for several stages of backbones, which share the same structures and parameters, but have various dilation rates $\alpha^{\mathrm{m}} = \{\alpha_i^{\mathrm{m}}, \ldots, \alpha_{n^{\mathrm{m}}}^{\mathrm{m}}\}$. Taking the $4^{\mathrm{th}}$ and $5^{\mathrm{th}}$ stages of backbone as an example, we first replicate the original $4^{\mathrm{th}}$ stage $n^{\mathrm{m}}$ times with various dilation rates, and for each output feature map repeat the replicating operation in the $5^{\mathrm{th}}$ stage. Finally, MRRP output $(n^{\mathrm{s}})^{(n^{\mathrm{m}})}$ feature maps in total, where $n^{\mathrm{s}}$ is the number of MRRP stages.

The proposed MRRP is used in UWSOD framework by two ways. The direct way is to average all feature maps for the successive processes. For the second way, we use SSOPG to generate object proposals on each feature map separately and apply NMS to all object proposals together. In RoIPool layer, the top $n_{\mathrm{infer}}^{\mathrm{p}}$ object proposals are map to their own feature maps to extract proposal features. Although it does not directly using entire resampling pyramid, scale-invariant features are still distilled into CNN by optimization of the shared parameters.

## 4 Quantitative evaluations

### 4.1 Datasets

We evaluate the proposed design principles on PASCAL VOC 2007, 2012 [86] and MS COCO [87], which are widely-used benchmark datasets. PASCAL VOC 2007 consists of $5,011$ *trainval* images,

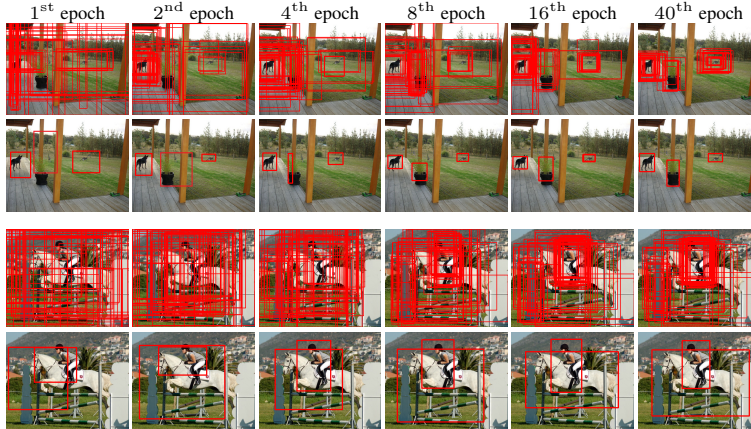

Figure 3: Visualize of the proposed SSOPG. The first and third rows show the top-50 object proposals generated by SSOPG. The second and forth rows show the pseudo-ground-truth proposal boxes.

Table 1: Ablation study on the VOC 2007 test set in terms of mAP and CorLoc.

| | SSOPG | SWBBFT | MRRP | CorLoc (%) | mAP (%) |
|---|---|---|---|---|---|
| ContextLocNet [29] | – | – | – | 55.1 | 36.3 |
| a | – | – | – | 55.7 | 36.7 |
| b | $n^p_{train}=1024, n^p_{infer}=1024$ | – | – | 50.8 | 32.1 |
| c | $n^p_{train}=2048, n^p_{infer}=2048$ | – | – | 50.9 | 32.3 |
| d | $n^p_{train}=4096, n^p_{infer}=4096$ | – | – | 50.8 | 32.1 |
| e | $n^p_{train}=2048, n^p_{infer}=2048$ | $n^f=3, \lambda^f=\{0.30,0.40,0.50\}$ | – | 60.5 | 41.8 |
| f | $n^p_{train}=2048, n^p_{infer}=2048$ | $n^f=4, \lambda^f=\{0.35,0.40,0.45,0.50\}$ | – | 60.8 | 42.1 |
| g | $n^p_{train}=2048, n^p_{infer}=2048$ | $n^f=5, \lambda^f=\{0.30,0.35,0.40,0.45,0.50\}$ | – | 60.5 | 42.0 |
| h | $n^p_{train}=2048, n^p_{infer}=2048$ | $n^f=4, \lambda^f=\{0.35,0.40,0.45,0.50\}$ | $n^m=2, \alpha^m=\{1,2\}$ | 62.4 | 43.3 |
| i | $n^p_{train}=2048, n^p_{infer}=2048$ | $n^f=4, \lambda^f=\{0.35,0.40,0.45,0.50\}$ | $n^m=3, \alpha^m=\{1,2,4\}$ | 62.6 | **44.0** |
| j | $n^p_{train}=2048, n^p_{infer}=2048$ | $n^f=4, \lambda^f=\{0.35,0.40,0.45,0.50\}$ | $n^m=3, \alpha^m=\{1,2,4\}^*$ | **63.0** | **44.0** |
| k | $n^p_{train}=2048, n^p_{infer}=2048$ | $n^f=4, \lambda^f=\{0.35,0.40,0.45,0.50\}$ | $n^m=4, \alpha^m=\{1,2,4,8\}$ | 62.8 | 43.6 |

and $4,092$ *test* images over 20 categories. PASCAL VOC 2012 consists of $11,540$ *trainval* images, and $10,991$ *test* images over 20 categories. Following the standard settings of WSOD, we use *trainval* set with only image-level labels for training. We also evaluate our approach on MS COCO, which is consists of 80 object categories. Our experiments use 118k *training* set with image-level labels for training, and 5k *validation* set for testing. Only image-level annotations are used in training.

## 4.2 Evaluation Protocol

Two protocols are used for evaluation: mean Average Precision (*m*AP) and CorLoc. The *m*AP follows standard PASCAL VOC protocol to report the *m*AP at $50\%$ Intersection-over-Union (IoU) of the detected boxes with the ground-truth ones. CorLoc quantifies the localization performance by the percentage of images that contain at least one object instance with at least $50\%$ overlapped to the ground-truth. For PASCAL VOC, we evaluate CorLoc and *m*AP on *trainval* and *testing*, respectively. For MS COCO, we report standard COCO metrics, including AP at different IoU thresholds.

## 4.3 Implementation details

We use VGG16 and WS-ResNet [28] backbones, which is initialized with the weights pre-trained on ImageNet ILSVRC [88]. We use synchronized SGD training on 4 GPUs. A mini-batch involves 1 images per GPU. We use a step learning rate decay schema with decay weight of $0.1$ and step size of $140,000$ iterations. The total number of training iterations is $200,000$. We adopt $2\times$ training schedules for MS COCO. In the multi-scale setting, we use scales range from $480$ to $1216$ with stride 32. To improve the robustness, we randomly adjust the exposure and saturation of the images by up to a factor of $1.5$ in the HSV space. A random crop with $0.9$ of the size of the original images is applied. We freeze all pre-trained convolutional layers in backbones unless specified otherwise. The test scores are the average of scales of $\{480, 576, 672, 768, 864, 960, 1056, 1152\}$ and flips. Detection results are post-processed by NMS with threshold of $0.5$. We use the following parameter settings

Table 2: Comparison with the state-of-the-art methods on PASCAL VOC 2007, 2012 and MS COCO.

| Method | | Bakbone | PASCAL VOC 2007 | | PASCAL VOC 2012 | | MS COCO Avg. Precision, IoU: | | |
|---|---|---|---|---|---|---|---|---|---|
| | | | mAP (%) | CorLoc (%) | mAP (%) | CorLoc (%) | 0.5:0.95 | 0.5 | 0.75 |
| WSOD with external object proposal modules or additional data | | | | | | | | | |
| Multi-Fold MIL | [6] | AlexNet | 30.2 | 52.0 | – | – | – | – | – |
| WSDDN | [7] | VGG16 | 34.8 | 53.5 | – | – | 9.5 | 19.2 | 8.2 |
| ContextLocNet | [29] | VGG-F | 36.3 | 55.1 | 35.3 | 54.8 | 11.1 | 22.1 | 10.7 |
| WCCN | [64] | VGG16 | 42.8 | 56.7 | 37.9 | – | – | – | – |
| Jie *et al.* | [34] | VGG16 | 41.7 | 56.1 | 38.3 | 58.8 | – | – | – |
| TST | [56] | AlexNet | 33.8 | 59.5 | – | – | – | – | – |
| SGWSOD | [78] | VGG16 | 43.5 | 62.9 | 39.6 | 62.9 | – | – | – |
| TS$^2$C | [32] | VGG16 | 44.3 | 61.0 | 40.0 | 64.4 | – | – | – |
| CSC C5 | [31] | VGG16 | 43.0 | 62.2 | 37.1 | 61.4 | 12.9 | 23.8 | 13.2 |
| WS-JDS | [48] | VGG16 | 45.6 | 64.5 | 39.1 | 63.5 | – | – | – |
| Oquab *et al.* | [79] | AlexNet | – | – | 11.7 | – | – | – | – |
| OICR | [8] | VGG16 | 41.2 | 60.6 | 37.9 | 62.1 | – | – | – |
| K-EM | [80] | VGG16 | 46.1 | 65.0 | – | – | – | – | – |
| MELM | [41] | VGG16 | 47.3 | 61.4 | 42.4 | – | – | – | – |
| ZLDN | [38] | VGG16 | 47.6 | 61.2 | 42.9 | 61.5 | – | – | – |
| GAL-fWSD512 | [47] | VGG16 | 47.5 | 66.1 | – | – | – | – | – |
| ML-LocNet | [42] | VGG16 | 48.4 | 67.0 | 42.2 | 66.3 | – | 16.2 | – |
| WSRPN | [11] | VGG16 | 45.3 | 63.8 | 40.8 | 64.9 | – | – | – |
| PCL | [36] | VGG16 | 43.5 | – | – | – | 8.5 | 19.4 | – |
| Kosugi *et al.* | [37] | VGG16 | 47.6 | 66.7 | 43.4 | 66.7 | – | – | – |
| C-MIL | [43] | VGG16 | 50.5 | 65.0 | 46.7 | 67.4 | – | – | – |
| Pred Net | [44] | VGG16 | 52.9 | 70.9 | 48.4 | 69.5 | – | – | – |
| OICR W-RPN | [12] | VGG16 | 46.9 | – | 43.2 | 67.5 | – | – | – |
| SDCN | [49] | VGG16 | 48.3 | 66.8 | 43.5 | 67.9 | – | – | – |
| Sona *et al.* | [81] | VGG16 | 45.4 | – | – | – | – | – | – |
| WSOD$^2$ | [18] | VGG16 | 53.6 | 69.5 | 47.2 | **71.9** | 10.8 | 22.7 | – |
| OICR+GAM+REG | [35] | VGG16 | 48.6 | 66.8 | – | – | – | – | – |
| C-MIDN | [50] | VGG16 | 52.6 | 68.7 | 50.2 | 71.2 | 9.6 | 21.4 | – |
| OIM+IR | [39] | VGG16 | 50.1 | 67.2 | 45.3 | 67.1 | – | – | – |
| Ren *et al.* | [20] | VGG16 | **54.9** | 68.8 | **52.1** | 70.9 | 12.4 | 25.8 | 10.5 |
| Zeni *et al.* | [82] | VGG16 | 49.7 | 65.7 | – | 66.3 | – | – | – |
| PG-PS | [13] | VGG16 | 51.1 | **69.2** | 48.3 | 68.7 | – | 20.7 | – |
| WSOD without external object proposal modules or additional data | | | | | | | | | |
| Shi *et al.* | [83] | – | – | 36.2 | – | – | – | – | – |
| Beam Search | [62] | VGG16 | 25.7 | – | 26.5 | – | – | – | – |
| OM+MIL | [33] | AlexNet | 23.4 | 41.2 | 29.1 | – | – | – | – |
| OPG | [84] | VGG16 | 28.8 | 43.5 | – | – | – | – | – |
| SPAM-CAM | [63] | VGG16 | 27.5 | – | – | – | – | – | – |
| UWSOD | | VGG16 | 44.0 | 63.0 | 45.1 | 65.2 | 2.5 | 9.3 | 1.1 |
| | | WSR18 | **45.0** | **63.8** | **46.2** | **65.7** | 3.1 | 10.1 | 1.4 |
| FSOD | | | | | | | | | |
| Fast RCNN | [85] | VGG16 | 66.9 | – | 65.7 | – | 18.9 | 38.6 | – |
| Faster RCNN | [2] | VGG16 | 69.9 | – | 67.0 | – | 21.2 | 41.5 | – |
| WSOD with Cls-agnostic GT-bbox Known | | | | | | | | | |
| OICR + GAM + REG[35] | | VGG16 | 54.3 | 81.3 | 53.9 | 82.1 | 13.7 | 27.1 | 12.5 |
| Ren *et al.* | [20] | VGG16 | 62.2 | 87.1 | 62.1 | 88.9 | 14.1 | 28.9 | 12.7 |
| UWSOD | | VGG16 | 67.7 | **93.3** | 65.3 | 91.1 | **15.3** | **32.4** | **12.8** |
| | | WSR18 | **69.7** | 92.5 | **66.1** | **92.3** | 13.7 | 27.9 | 12.5 |

in all the experiments, unless specified otherwise. We set the labeling threshold $\lambda^{obn}$ and $\lambda^p$ to 0.5 and 0.7, respectively. For SWBBFT, We se the number of fine-tune branches $n^f$ to 4, and $\lambda^f$ to $\{0.3, 0.4, 0.5, 0.6\}$. We apply MRRP on the last stage of backbone with $n^m = 3$ and $\alpha^m = \{1, 2, 4\}$.

## 4.4 Ablation study

We validate the contribution of each design components on PASCAL VOC 2007 in Tab. 1. We use ContextLocNet [29] as our baseline, which is widely used in recent WSOD methods. Our implementation of WSDDN in row (a) has superior performance, which may due to larger mini-batch

size and epochs. For rows (b-d), we report the results of applying SSOPG with various parameters, which show competitive performance compared to the original (a). We observe that decreasing object proposals generated by SSOPG drops the performance, as the recall rate is not sufficient enough to support accuracy object mining in WSOD. Rows (e-g) show the results of applying SWBBFT, which provides performance boosting from row (c). The benefits are mainly from: First, the proposed step-wise learning paradigm progressively selects high-confidence object proposals as positive samples for refining. Second, each refinement branch has bounding-box regressor to refine bounding-box coordinates step-wisely. Rows (h-k) show that the proposed MRRP further improves both the localization and detection performance. It demonstrates that MRRP remedies the scale variations issue by the in-network feature hierarchy. MRRP in row (j) maps object proposals to their own pyramid level, which also achieves competitive results compared to row (i), which averages all feature maps. As shown in Fig. 3, the initial proposals are spatially scattered. With more training epochs, SSOPG rapidly learns to generate proposals that are clustered around object gradually.

### 4.5   Comparison with the state of the arts

We compared our proposed method with previous methods based on a single backbone. In Tab. 2, we first compare the results on Pascal VOC 2007 in terms of $m$AP and CorLoc. As shown in the bottom of Tab. 2, the proposed UWSOD significantly outperforms the state-of-the-art methods that do not use external modules or additional data. This indicates the efficiency of UWSOD. Compared to other methods with external object proposals or data, UWSOD also achieves competitive results. We also report the performance on Pascal VOC 2012 in Tab. 2. UWSOD consistently outperforms other self-contained methods and achieves new state-of-the-art results. The benefits are mainly from effectively learning object proposal generation, bounding-box fine-tuning and scale-invariant features. The last column of Tab. 2 shows the results on MS COCO. The low performance is mainly due to that COCO dataset has more complex scenes and larger category set.

### 4.6   Upper-bound performance analysis

As WSOD is upper bounded in its capacity for object localization, we further analyze the upper-bound performance of WSOD methods with ground-truth bounding boxes (GT-bbox Known).The upper bound of recent state-of-the-art methods, *e.g.*, [35, 20], are Fast R-CNN [85], as they all heavily rely on external object proposal algorithms. Different to them, UWSOD learns to generate object proposals and has upper-bound performance as with Faster R-CNN [2]. We further introduce a regression-disentangled learning setting to decouple proposal classification and regression tasks. In detail, we remove the class annotation from the ground-truth bounding-box labels during training, termed Cls-agnostic GT-bbox Known. Thus, WSOD still need to capture coarse object location, while predicted boxes have ground-truth regression supervision to fine-tune themselves. As shown in the bottom part of Tab. 2, the performance of UWSOD approaches its upper bound of GT-bbox Known, which also demonstrates that UWSOD has the fully-supervised-level capacity. We find that the accurate bounding-box localization is one of the main obstacles to reduce the performance gap between UWSOD and its fully supervised counterpart.

## 5   Conclusion

In this paper, we propose a unified WSOD framework, termed UWSOD, to develop a high-capacity general detection model with only image-level labels, which is self-contained and does not require external modules or additional supervision. We focus on three important components, *i.e.*, object proposal generation, bounding-box fine-tuning and scale-invariant features. Extensive experiments on PASCAL VOC and MS COCO show that the proposed method competitive results with the state-of-the-art WSOD methods while not requiring external modules or additional supervision. We also demonstrate that UWSOD obtains higher upper-bound performance than other WSOD methods and has fully-supervised-level capacity, which brings larger potential to reduce the performance gap between WSOD and FSOD methods.

## Broader Impact

WSOD aims at leveraging weakly supervised learning to train object detectors, which significantly reduces the human labelling effort. Therefore, WSOD has the potential of handling thousands of real-world categories and taking advantage of large-scale weak annotations. In this work, we develop a unified high-capacity generic object detector with image-level labels, termed UWSOD, and exploit three important components, *i.e.*, object proposal generation, bounding-box fine-tuning and scale-invariant features, all of which are rarely touched in WSOD before. Our method has both practical and methodological contributions to facilitate the development of this area.

- For the academia, the superior capacity of our method demonstrates that WSOD has potential to achieve competitive results compared to FSOD.
- For the industry, the proposed UWSOD method enables to utilize the image-level annotations cheaply available on the Internet to learn detectors.
- For the community, we hope to give new insight into other tasks under weak supervision to achieve promising performance.

## Acknowledgments and Disclosure of Funding

This work is supported by the National Natural Science Foundation of China (No.U1705262, No.61772443, No.61572410, No.61802324 and No.61702136), National Key R&D Program (No.2017YFC0113000, and No.2016YFB1001503), Key R&D Program of Jiangxi Province (No.20171ACH80022) and Natural Science Foundation of Guangdong Province in China (No.2019B1515120049), National key research and development plan project (No.2018YFC0830105, and No.2018YFC0830100).

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
