[Supplementary Material]

# Supplementary material

## Additional results on PASCAL VOC

In Tab. S1 and Tab. S2, we report the per-category detection APs and localization CorLocs on the *test* and *trainval* splits of PASCAL VOC 2007, respectively. Compared to other WSOD methods without external modules or data, we observe: (1) Our UWSOD outperforms all others on most categories (19 categories for APs and CorLocs). (2) The largest improvement in terms of AP is from category *cow*, which has gains of 37.9. (3) The performance of hard categories in our UWSOD, *i.e.*, *boat*, *chair* and *plant*, still surpass other methods by 10.3, 0.3 and 5.0 AP points, respectively. With class-agnostic ground-truth bounding-box known, we observe that the per-category performance of proposed UWSOD is competitive to that of Fast/Faster RCNN.

Table S1: Comparison with the state-of-the-art methods on PASCAL VOC 2007 in terms of AP (%) on *test*.

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

Figure S1: Proposal objectness score maps of different resampling rate $\alpha^m$ in MRRP on PASCAL VOC 2007 test.

Table S2: Comparison with the state-of-the-art methods on PASCAL VOC 2007 in terms of CorLoc (%) on *trainval*.

| Method | | Backbone | aero | bicy | bird | boa | bot | bus | car | cat | cha | cow | dtab | dog | hors | mbik | pers | plnt | she | sofa | trai | tv | Av. |
|---|---|---|---|---|---|---|---|---|---|---|---|---|---|---|---|---|---|---|---|---|---|---|---|
| WSOD with external object proposal modules or additional data | | | | | | | | | | | | | | | | | | | | | | | |
| Multi-Fold MIL | [6] | AlexNet | 65.3 | 55.0 | 52.4 | 48.3 | 18.2 | 66.4 | 77.8 | 35.6 | 26.5 | 67.0 | 46.9 | 48.4 | 70.5 | 69.1 | 35.2 | 35.2 | 69.6 | 43.4 | 64.6 | 43.7 | 52.0 |
| WSDDN | [7] | VGG16 | 65.1 | 58.8 | 58.5 | 33.1 | 39.8 | 68.3 | 60.2 | 59.6 | 34.8 | 64.5 | 30.5 | 43.0 | 56.8 | 82.4 | 25.5 | 41.6 | 61.5 | 55.9 | 65.9 | 63.7 | 53.5 |
| ContextLocNet | [29] | VGG-F | 83.3 | 68.6 | 54.7 | 23.4 | 18.3 | 73.6 | 74.1 | 54.1 | 8.6 | 65.1 | 47.1 | 59.5 | 67.0 | 83.5 | 35.3 | 39.9 | 67.0 | 49.7 | 63.5 | 65.2 | 55.1 |
| WCCN | [64] | VGG16 | 83.9 | 72.8 | 64.5 | 44.1 | 40.1 | 65.7 | 82.5 | 58.9 | 33.7 | 72.5 | 25.6 | 53.7 | 67.4 | 77.4 | 26.8 | 49.1 | 68.1 | 27.9 | 64.5 | 55.7 | 56.7 |
| Jie et al.'17 | [34] | VGG16 | 72.7 | 55.3 | 53.0 | 27.8 | 35.2 | 68.6 | 81.9 | 60.7 | 11.6 | 71.6 | 29.7 | 54.3 | 64.3 | 88.2 | 22.2 | 53.7 | 72.2 | 52.6 | 68.9 | 75.5 | 56.1 |
| TST | [56] | AlexNet | – | – | – | – | – | – | – | – | – | – | – | – | – | – | – | – | – | – | – | – | 59.5 |
| SGWSOD | [78] | VGG16 | 71.0 | 76.5 | 54.9 | 49.7 | 54.1 | 78.0 | 87.4 | 68.8 | 32.4 | 75.2 | 29.5 | 58.0 | 67.3 | 84.5 | 41.5 | 49.0 | 78.1 | 60.3 | 62.8 | 78.9 | 62.9 |
| TS$^2$C | [32] | VGG16 | 84.2 | 74.1 | 61.3 | 52.1 | 32.1 | 76.7 | 82.9 | 66.6 | 42.3 | 70.6 | 39.5 | 57.0 | 61.2 | 88.4 | 9.3 | 54.6 | 72.2 | 60.0 | 65.0 | 70.3 | 61.0 |
| CSC C5 | [31] | VGG16 | 76.1 | 75.3 | 61.8 | 42.0 | 54.1 | 74.7 | 78.8 | 67.4 | 32.8 | 73.1 | 46.5 | 59.9 | 37.6 | 78.0 | 56.0 | 42.5 | 71.9 | 67.3 | 82.4 | 65.6 | 62.2 |
| WS-JDS | [48] | VGG16 | 82.9 | 74.0 | 73.4 | 47.1 | 60.9 | 80.4 | 77.5 | 78.8 | 18.6 | 70.0 | 56.7 | 67.0 | 64.5 | 84.0 | 47.0 | 50.1 | 71.9 | 57.6 | 83.3 | 43.5 | 64.5 |
| OICR | [8] | VGG16 | 81.7 | 80.4 | 48.7 | 49.5 | 32.8 | 81.7 | 85.4 | 40.1 | 40.6 | 79.5 | 35.7 | 33.7 | 60.5 | 88.8 | 21.8 | 57.9 | 76.3 | 59.9 | 75.3 | 81.4 | 60.6 |
| K-EM | [80] | VGG16 | 79.8 | 77.8 | 66.7 | 50.3 | 57.0 | 80.1 | 89.9 | 71.5 | 29.9 | 75.9 | 30.5 | 58.9 | 73.2 | 90.2 | 25.4 | 51.8 | 80.2 | 60.3 | 72.4 | 78.9 | 65.0 |
| MELM | [41] | VGG16 | – | – | – | – | – | – | – | – | – | – | – | – | – | – | – | – | – | – | – | – | 61.4 |
| ZLDN | [38] | VGG16 | 74.0 | 77.8 | 65.2 | 37.0 | 46.7 | 75.8 | 83.7 | 58.8 | 17.5 | 73.1 | 49.0 | 51.3 | 76.7 | 87.4 | 30.6 | 47.8 | 75.0 | 62.5 | 64.8 | 68.8 | 61.2 |
| GAL-fWSD512 | [47] | VGG16 | 78.6 | 81.9 | 63.6 | 40.3 | 48.8 | 80.7 | 85.3 | 76.3 | 30.3 | 78.0 | 54.5 | 65.3 | 48.4 | 86.5 | 56.3 | 46.9 | 76.0 | 68.1 | 83.9 | 73.1 | 66.1 |
| ML-LocNet | [42] | VGG16 | 78.6 | 82.3 | 68.2 | 42.0 | 53.3 | 78.5 | 88.5 | 70.3 | 36.4 | 70.2 | 60.5 | 58.0 | 80.5 | 88.2 | 38.8 | 59.2 | 75.0 | 69.0 | 78.2 | 64.5 | 67.0 |
| WSRPN | [11] | VGG16 | 77.5 | 81.2 | 55.3 | 19.7 | 44.3 | 80.2 | 86.6 | 69.5 | 10.1 | 87.7 | 68.4 | 52.1 | 84.4 | 91.6 | 57.4 | 63.4 | 77.3 | 58.1 | 57.0 | 53.8 | 63.8 |
| Kosugi et al. | [37] | VGG16 | 85.5 | 79.6 | 68.1 | 55.1 | 33.6 | 83.5 | 83.1 | 78.5 | 42.7 | 79.8 | 37.8 | 61.5 | 74.4 | 88.6 | 32.6 | 55.7 | 77.9 | 63.7 | 78.4 | 74.1 | 66.7 |
| C-MIL | [43] | VGG16 | – | – | – | – | – | – | – | – | – | – | – | – | – | – | – | – | – | – | – | – | 65.0 |
| Pred Net | [44] | VGG16 | 88.6 | 86.3 | 71.8 | 53.4 | 51.2 | 87.6 | 89.0 | 65.3 | 33.2 | 86.6 | 58.8 | 65.9 | 87.7 | 93.3 | 30.9 | 58.9 | 83.4 | 67.8 | 78.7 | 80.2 | 70.9 |
| OICR W-RPN | [12] | VGG16 | - | - | - | - | - | - | - | - | - | - | - | - | - | - | - | - | - | - | - | - | 66.5 |
| SDCN | [49] | VGG16 | 85.8 | 83.1 | 56.2 | 58.5 | 44.7 | 80.2 | 85.0 | 77.9 | 29.6 | 78.8 | 53.6 | 74.2 | 73.1 | 88.4 | 18.2 | 57.5 | 74.2 | 60.8 | 76.1 | 79.2 | 66.8 |
| WSOD$^2$ | [18] | VGG16 | 87.1 | 80.0 | 74.8 | 60.1 | 36.6 | 79.2 | 83.8 | 70.6 | 43.5 | 88.4 | 46.0 | 74.7 | 87.4 | 90.8 | 44.2 | 52.4 | 81.4 | 61.8 | 67.7 | 79.9 | 69.5 |
| OICR+GAM+REG | [35] | VGG16 | 81.7 | 81.2 | 58.9 | 54.3 | 37.8 | 83.2 | 86.2 | 77.0 | 42.1 | 83.6 | 51.3 | 44.9 | 78.2 | 90.8 | 20.5 | 56.8 | 74.2 | 66.1 | 81.0 | 86.0 | 66.8 |
| C-MIDN | [50] | VGG16 | - | - | - | - | - | - | - | - | - | - | - | - | - | - | - | - | - | - | - | - | 68.7 |
| OIM+IR | [39] | VGG16 | - | - | - | - | - | - | - | - | - | - | - | - | - | - | - | - | - | - | - | - | 67.2 |
| Ren et al. | [20] | VGG16 | 87.5 | 82.4 | 76.0 | 58.0 | 44.7 | 82.2 | 87.5 | 71.2 | 49.1 | 81.5 | 51.7 | 53.3 | 71.4 | 92.8 | 38.2 | 52.8 | 79.4 | 61.0 | 78.3 | 76.0 | 68.8 |
| Zeni et al. | [82] | VGG16 | 86.7 | 73.3 | 72.4 | 55.3 | 46.9 | 83.2 | 87.5 | 64.5 | 44.6 | 76.7 | 46.4 | 70.9 | 67.0 | 88.0 | 9.6 | 56.4 | 69.1 | 52.4 | 79.8 | 82.8 | 65.7 |
| PG-PS | [13] | VGG16 | 85.4 | 80.4 | 69.1 | 58.0 | 35.9 | 82.7 | 86.7 | 82.6 | 45.5 | 84.9 | 44.1 | 80.2 | 84.0 | 89.2 | 12.3 | 55.7 | 79.4 | 63.4 | 82.1 | 82.1 | 69.2 |
| WSOD without external object proposal modules or additional data | | | | | | | | | | | | | | | | | | | | | | | |
| Shi et al. | [83] | – | 67.3 | 54.4 | 34.3 | 17.8 | 1.3 | 46.6 | 60.7 | 68.9 | 2.5 | 32.4 | 16.2 | 58.9 | 51.5 | 64.6 | 18.2 | 3.1 | 20.9 | 34.7 | 63.4 | 5.9 | 36.2 |
| OM + MIL | [33] | AlexNet | 64.3 | 54.3 | 42.7 | 22.7 | 34.4 | 58.1 | 74.3 | 36.2 | 24.3 | 50.4 | 11.0 | 29.2 | 50.5 | 66.1 | 11.3 | 42.9 | 39.6 | 18.3 | 54.0 | 39.8 | 41.2 |
| OPG | [84] | VGG16 | 57.1 | 43.2 | 53.9 | 23.8 | 12.3 | 47.9 | 48.8 | 69.1 | 16.6 | 47.5 | 39.0 | 61.3 | 54.7 | 60.8 | 32.1 | 22.0 | 49.0 | 44.1 | 59.4 | 27.7 | 43.5 |
| UWSOD | | VGG16 | 77.8 | 85.8 | 66.0 | 56.0 | 39.1 | 74.2 | 91.4 | 41.4 | 30.3 | 81.9 | 33.0 | 78.9 | 90.5 | 85.6 | 7.6 | 46.4 | 68.8 | 67.0 | 76.1 | 61.7 | 63.0 |
| UWSOD | | WSR18 | 80.4 | 85.3 | 79.4 | 42.0 | 65.5 | 78.4 | 90.7 | 49.7 | 18.8 | 73.9 | 48.5 | 63.1 | 87.8 | 90.8 | 37.4 | 47.4 | 77.1 | 54.1 | 81.9 | 23.4 | 63.8 |
| WSOD with Cls-agnostic GT-bbox Known | | | | | | | | | | | | | | | | | | | | | | | |
| OICR + GAM + REG | [35] | VGG16 | 91.9 | 82.7 | 87.2 | 65.4 | 69.7 | 92.7 | 92.2 | 82.8 | 45.9 | 88.4 | 80.4 | 84.7 | 89.9 | 90.8 | 79.8 | 59.4 | 89.5 | 79.2 | 90.5 | 83.5 | 81.3 |
| Ren et al. | [20] | VGG16 | 95.3 | 94.5 | 93.0 | 83.1 | 68.5 | 92.8 | 91.8 | 93.0 | 69.8 | 86.4 | 71.2 | 89.1 | 92.9 | 96.3 | 88.9 | 75.4 | 93.8 | 82.6 | 92.4 | 89.4 | 87.1 |
| UWSOD | | VGG16 | 98.4 | 96.8 | 96.6 | 98.0 | 81.9 | 96.6 | 96.7 | 97.1 | 78.2 | 98.6 | 82.5 | 96.3 | 97.3 | 95.2 | 95.1 | 80.3 | 97.9 | 91.5 | 98.5 | 92.1 | 93.3 |
| UWSOD | | WSR18 | 99.1 | 96.6 | 97.8 | 91.4 | 81.3 | 93.8 | 95.5 | 94.5 | 70.1 | 95.7 | 90.7 | 97.0 | 98.6 | 97.5 | 94.7 | 81.2 | 100.0 | 91.0 | 97.6 | 86.0 | 92.5 |

## Additional analysis of MRRP

In Fig. S1, we show the proposal objectness score maps of different resampling rates in MRRP with $n^{\mathrm{m}} = 3$ and $\alpha^{\mathrm{m}} = \{1, 2, 4\}$. We accumulate the objectness scores of proposals with the same height and width from PASCAL VOC 2007 *test*. Then we map the accumulated objectness scores to the jet colourmap. We find that different resampling rates tend to favour different sizes of proposals. With large resampling rates, the number of the large proposals with high objectness scores is much more than that of small ones. And small resampling rates, *e.g.*, $\alpha^{\mathrm{m}} = 1$, predict more small proposals with high objectness scores. It demonstrates that our MRRP uses the in-network feature hierarchy to handle large scale variation.