[Reviews · NeurIPS 2020]

Review 1

Summary and Contributions: This paper proposes a new approach for the weakly-supervised object detection (using only image-level annotations no bounding box). Using their approach, they obtain state-of-the-art results on PASCAL VOC and MS COCO in the WSOD setting.

Strengths: + Paper proposes three main components in the WSOD setting: 1)  Anchor-based self-supervised proposal generator network 2)  Step-wise finetuning of the bounding box 3)  Multi-rate resampling pyramid for hierarchical features to handle scale variations + In table 1, they perform detailed ablation study to show the importance of each component. + They obtain state-of-the-art results compared to all the recent approaches on all the three datasets (PASCAL VOC 2007, PASCAL VOC 2012, MS COCO). + Paper is well written and easy to understand

Weaknesses: - It would be great if authors can mention some failure cases of their approach. Especially for the MS COCO, it would be good to know how their approach performs for the smaller objects. - Also, it would be nice to have some more qualitative results of their approach.  - Also, it would be nice to see the performance of their approach if they uses Selective Search or Edge Boxes.

Correctness: All the claims seem correct.

Clarity: Paper is well written.

Relation to Prior Work: The paper does a good job in discussing the prior work.

Reproducibility: Yes

Additional Feedback: Post rebuttal: The results are impressive but it lacks detailed explanantion and insights why they are getting such gains compared to previous approaches as pointed by R4. This paper would be much stronger if they include such discussions. Also, it turns out their learned proposals has a little contribution as replacing it with edge box and selective search have only a small drop in performance. So, overall it is not clear what is the exact cause of such big boost in performance. Considering all these, I am lowering my score to 6.


Review 2

Summary and Contributions: This paper presents a new method for weakly supervised object detection (WSOD). It is based on 3 parts: an anchor-based self-supervised proposal generator, a step-wise bounding-box refinement, and a re-sampling pyramid. The method outperforms all previous WSOD approaches by a margin and approaches the performance of fully supervised detectors.

Strengths: - The obtained performance is impressive, approaching fully supervised methods performance - WSOD is a very important task because it can produce powerful detectors without the need of training with bounding-box annotations - The ablation study shows that each component of the method helps to improve performance.

Weaknesses: - The proposed method is quite complex and its presentation is in some points difficult to follow. Figure 2 helps to give a global understanding of the model, but a richer caption with a brief description of each part could help. - The performance of the method stems from multiple IoU thresholds in SWBBFT and multiple sampling scales in MRRP. These approaches seem to add a relevant computational cost. The authors should comment on that.

Correctness: How the hyper-parameters are chosen without a validation set? This is a common problem on all weakly supervised approaches, but this does not mean that it should not be considered. In particular, as the proposed method has many components and therefore many hyper-parameters, the increased performance could be partially due to a better tuning of more hyper-parameters.

Clarity: The presentation of the method is difficult to understand at the first read. It can be because the proposed method is quite complex, but I also think that some additional explanations and more detailed figures could help.

Relation to Prior Work: In related work authors present and relate the most relevant previous work to the proposed approach.

Reproducibility: Yes

Additional Feedback: -------------after rebuttal comment----------------------------- Considering other reviewers comments as well as authors rebuttal I reduced my score to 7 mostly because the presentation of the method is not very clear and it is difficult to fully understand the reason for such good performance.


Review 3

Summary and Contributions: This work aims to tackle the weakly supervised object detection task only using image-level labels as supervision. The proposed approach does not rely on offline generated proposals and can be leaned in an end-to-end manner. Extensive experiments on VOC and COCO demonstrate the effectiveness of the proposed method.

Strengths: 1. Using the learnable proposals is meaningful to the WSOD research area. 2. The proposed approach makes a significant improvement over previous approaches.

Weaknesses: I am not very clear for section 3.2, especially for the content from line 181 to line 196. How to construct T_i^{obn}, T_i^{p}, \bar{B}_i^{obn}, \bar{B}_i^{p}? Additionally, how to set lamda^{obn} and lamda^{p}? are these two hyperparameters sensitive to the final detection performance? In table 1, most experiments are based on n^p_{train}=2048. From supp, I notice that MCG and Edgebox proposals are with better Recall-IOU curves compared to SSOPG. So, if the proposal branch is replaced by SS or MCG, can the framework achieve better performance?

Correctness: Yes

Clarity: Some parts need to be improved.

Relation to Prior Work: Yes

Reproducibility: Yes

Additional Feedback: In general, I would like to accept this work if the weaknesses are well addressed. --------- My previous rating score mainly comes from the super performance and the end-to-end framework. After carefully checking the comments from R4, I agree that this submission lacks plausible explanation to the super performance. Regardiing this, I would like to reduce my score to 6.


Review 4

Summary and Contributions: This paper proposes a unified WSOD framework, termed UWSOD. There are three main components, i.e., object proposal generation, bounding-box fine-tuning and scale-invariant features. The authors first use an anchor-based self-supervised proposal generator to hypothesize object locations, which is trained end-to-end with supervision created by UWSOD for both objectness classification and regression. Second, a step-wise bounding-box fine-tuning model is developed to refine both detection scores and coordinates by progressively select high-confidence object proposals as positive samples. The goal is to bootstrap the quality of predicted bounding boxes. Third, a multi-rate resampling pyramid is constructed to aggregate multi-scale contextual information to handle scale variation in WSOD.

Strengths: +Studies an interesting yet challenging task in the vision community. + Good performance is shown in the tables.

Weaknesses: -I have to say the technical novelty of this work is not significant. All the three components considered by the authors have already been explored by existing works, e.g., [11] for the first component, [17] for the second component, and [72] for the third component. Although the implementation is not exactly the same as the existing works, they share very similar spirits. -In table 1, I observe that the SWBBFT model brings about 10 percent performance gain. It is not clear why the existing works, such as [17-20], cannot obtain such huge performance gain by using the similar network modules. -I cannot see why the method can work. To my best knowledge, the weak image labels cannot provide sufficient supervision to train a complicated network like the one proposed in this paper. It is not clear without meaningful proposals, how to train WSDDN. And without good WSDDN results, how to train SWBBFT and SSOPG.

Correctness: I do not think the proposed method can correctly address the WSOD problem.

Clarity: The paper fails to provide convincing insights for addressing the WSOD problem.

Relation to Prior Work: Yes, the difference is discussed in the Introduction.

Reproducibility: No

Additional Feedback: --------------------------------------update after rebuttal------------------------------------------ In the response file, the authors answered some of my questions. However, my concerns are not fully addressed. Specifically, I know there are some differences with [11,17,72], but I do not think the differences are significant. As the authors said in rebuttal “…while work in [72] focused on encoding traditional image descriptions in fully-supervised learning”. It is hard for me to agree that bring a technique in fully-supervised learning in weakly supervised learning is a significant innovation. In R4Q2, the authors mentioned the improvement from SWBBFT as compared to the vanilla instance refinement in OICR. However, my question is about how SWBBFT outperforms [17-20]. Ideas like “increasing quality of pseudo-ground truths and a good balance between positive and negative samples” have also been explored by these works and the authors did not clarify why they can obtain much better performance by using a similar idea. Finally, in R4Q3, the authors said “First, WSDDN formulates WSOD as multiple instance learning and captures the target object from a large set of proposals.” However, as SSOPG has not been trained at that time, how to generate the so-called “high-recall” proposals for WSDDN? Due to these issues, I think this work is not ready for publication.

[Author Response · NeurIPS 2020]

We sincerely thank all reviewers and ACs for their time and efforts. We also appreciate the positive comments on our
novelty, contributions and state-of-the-art performance, *e.g.*, "impressive,", "meaningful", "detailed ablation study"
and "well written and easy to understand". We have released code in GitHub anonymously, which can be searched by
querying for "UWSOD". Below we give our responses point-by-point.

**R1Q1 & R1Q2: More failure cases and qualitative results. The performance for the smaller objects in COCO.**

**A:** We will analyze some failure cases and qualitative results in the supplementary material as suggested. The $mAP$ of
the small object in MS COCO is only $2.2\%$, as it is more challenge to detect small object than the large one in WSOD.

**R1Q3 & R3Q2: The performance of the approach if they use Selective Search (SS), Edge Boxes (EB) or MCG.**

**A:** Replacing SSOPG with traditional SS and EB proposals decreases $1.5\%$ and $1.1\%$ $mAP$ on Pascal VOC 2007. And
MCG proposals improve the detection performance of UWSOD by $1.5\%$ $mAP$. However, MCG takes about $34.3$ s. per
image to generate proposals, which is time-consuming for practical applications. We will add the above results to paper.

**R2Q1: The proposed method is quite complex and its presentation is in some points difficult to follow.**

**A:** Thanks for your suggestions on presentation. We will revise the paper thoroughly under your suggestions.

**R2Q2: These approaches seem to add a relevant computational cost. The authors should comment on that.**

**A:** Our implemented WSDDN on VGG16 takes $439$ ms. per images in GTX 1080TI GPU and PASCAL VOC. When
we sequentially add SWBBFT, SSOPG and MRRP, the training times increase to $618$, $848$ and $1,581$ ms., respectively.

**R2Q3: How the hyper-parameters are chosen without a validation set?**

**A:** Most hyper-parameters are set based on the previous WSOD and FSOD methods. And the hyper-parameters of our
components are manually chosen by intuition and experience, which are kept the same for different dataset.

**R3Q1: How to set $T_i^{obn}$, $T_i^p$, $\bar{B}_i^{obn}$, $\bar{B}_i^p$, and $\lambda^{obn}$, $\lambda^p$? Are these two hyperparameters sensitive to performance?**

**A:** $T_i^{obn}$, $T_i^p$, $\bar{B}_i^{obn}$, $\bar{B}_i^p$ are the classification and regression targets for objecness detection branch and object proposal
generator, respectively. They are set based on IoU threshold between proposals and the corresponding pseudo-ground-
truths. As we stated in the Implementation details, we set the labelling threshold $\lambda^{obn}$ and $\lambda^p$ to $0.5$ and $0.7$, respectively.
And varying $\lambda^{obn}$ and $\lambda^p$ within a range of $[-0.1, 0.1]$ only decreases the performance by about $1.3 \sim 2.1\%$ $mAP$.

**R4Q1: All three components considered by the authors have already been explored by existing works[11,17,72].**

**A:** As we emphasized in the Introduction, there are significant differences between our method and the existing works.
In particular, WSOD methods in [11,17] relied on external object proposal algorithms or additional instance-level
supervision to learn detectors, while work in [72] focused on encoding traditional image descriptions in fully-supervised
learning. To our best knowledge, our work is the *first* to propose learnable object proposals (SSOPG) without external
modules or additional supervision and aggregate multi-scale in-network contextual information (MRRP) for WSOD
task. And SWBBFT improves instance refinement from a new perspective of the trade-off between precision and recall
requirements in different branches. Our method has both practical and methodological contributions to facilitate the
development of this area. Moreover, detaching detectors from external time-consume modules makes WSOD more
capable of handling thousands of real-world categories and taking advantage of large-scale weak annotations.

**R4Q2: Why [17-20] cannot obtain such huge performance gain by using similar network modules (SWBBFT)?**

**A:** Actually, careful designs of instance refinement module have shown significant improvement of performance in
many works. The instance refinement proposed in OICR [8] improves WSDDN by $6.4\%$ $mAP$. Recent methods
in [17,18,20,35] proposed various strategy to improve the instance refinement module, which achieved gains of
$5.4\%$, $7.0\%$, $8.9\%$ and $6.0\%$ $mAP$, respectively, compared to vanilla instance refinement in OICR. We attribute the
improvement from SWBBFT to a sequence of effective refinement branches of increasing quality of pseudo-ground-
truths and a well balance between positive and negative samples, which are neglected in the previous works.

**R4Q3: It is not clear without meaningful proposals, how to train WSDDN, SWBBFT and SSOPG?**

**A:** We further exhibit how our UWSOD works in the learning process. First, WSDDN formulates WSOD as multiple
instance learning and captures the target object from a large set of proposals. Therefore, we enforce SSOPG to
output redundancy object proposals to ensure high recall, which makes WSDDN capable of selecting positive samples.
Although the output proposals of initial SSOPG are messy, WSDDN still has high-probability of finding informative
proposals, which either contain discriminative object part or cover the entire object loosely. Second, with the output of
WSDDN, a sequence of effective refinements in SWBBFT bootstraps the quality of predicted bounding boxes, which
has been demonstrated to improve performance in many existing works [8,17-20,35-36]. Third, a self-supervised
learning is leveraged to train SSOPG with supervision distilled from SWBBFT, which in turn improves WSDDN results.

[Meta-Review · NeurIPS 2020]

This paper proposes a new framework for WSOD, with great performance (close to FSOD). Ablation results are provided, and the method is based on prior work but with important differences. Code has been shared. It is not fully clear why performance is such an improvement over prior WSOD, since prior methods have explored similar ideas (in different form). However, several expert reviewers agreed that the community would benefit from seeing this work.